# The effectiveness of group interpersonal synchrony in young autistic adults' work environment: A mixed methods RCT study protocol

**Tamar Dvir** ⓘ *, **Tal-Chen Rabinowitch, Cochavit Elefant**

University of Haifa, Faculty of Social Welfare & Health Sciences, School of Creative Arts Therapies, Haifa, Israel

* tamar.dvir@gmail.com

## Abstract

### Introduction

Few autistic adults are able to integrate successfully into the world of work given their difficulties adapting to the social and stressful aspects of work environments. Interpersonal synchrony, when two or more individuals share body movements or sensations, is a powerful force that consolidates human groups while promoting the ability to self-regulate and cooperate with others. The abilities to self-regulate and cooperate are crucial for maintaining a calm and productive work environment. This study protocol outlines research that aims to assess the effects of group interpersonal synchrony on prosociality and work-related stress of young autistic adults in their work environment.

### Methods and analysis

This mixed-methods randomized controlled trial will investigate two movement-based group synchronous and non-synchronous intervention conditions. The sample will be composed of young adults enrolled in an innovative Israeli program designed to integrate cognitively-abled 18- to 25-year-old autistic adults into the Israeli army work force. The movement-based intervention sessions will take place in groups of 10–14 participants, once a week for 10 weeks. Questionnaires, behavioral collaborative tasks and semi-structured interviews will be conducted. Quantitative data will be collected for each participant at three points of time: before and after the intervention period, and 17 weeks after the end of the intervention. Qualitative data will be collected after the intervention period in interviews with the participants.

### Discussion

Little is known about interventions that promote successful integration into social and stressful work environments. The findings are likely to shed new light on the use of group interpersonal synchrony in autistic individuals at work.

**Data Availability Statement:** Deidentified research data will be made publicly available when the study is completed and published.

**Funding:** The author(s) received no specific funding for this work.

**Competing interests:** The authors have declared that no competing interests exist.

## Trial registration

NCT05846308.

## Introduction

Individuals diagnosed with Autism Spectrum Disorder (ASD) have difficulties related to social interaction and communication, as well as repetitive and stereotyped interests [1]. According to the DSM-V [1] cognitively-abled autistic people are characterized by a high intellectual level and the least requirements for support. There are few cognitively-abled autistic adults who can successfully integrate into the world of work since they often manifest atypical behavior in social situations, find it hard to initiate social interactions, have limited interest in forming social relationships, find it challenging to cooperate with co-workers and often fail to regulate work-related stress [2–4]. The ability to work, according to the International Classification of Functioning, Disability and Health (World Health Organization, 2001) is considered to be a significant factor affecting individuals' health, quality of life, and wellbeing. This accounts for the growing interest in developing tailored interventions to promote the long-term integration of challenged populations into social and stressful work environments [5, 6].

A new wave of research in autism suggests that in addition to difficulties with social interactions and stressful environments, ASD is also characterized by fundamental differences in sensorimotor functioning including the delayed development of motor skills [7], abnormal gait [8], difficulty with balance [9], difficulties in coordinating movements that involve both sides of the body, and in controlling force and direction when throwing a ball [10]. These altered modes of perceiving and moving in the world (alone, as well as with other people) may help explain some of the key fundamental social impairments in autism and asynchronous social behavior in particular [11–15]. Asynchronous social behavior in autism (e.g., reduced eye contact, absence of well-timed co-regulation, no coherent engagement of mutual attention, no anticipation and no emotional build up in shared interactions) was first identified in a micro-analytic study of videos of an 11-month old infant who was diagnosed with ASD during her second year of life [16]. Stronger evidence for the association between atypical interpersonal synchrony and autism was recently described [13] in a study which also suggested that there may be a disruption in intrapersonal mechanisms such as atypical motor timing, and in interpersonal mechanisms such as atypical inter-individual coupling.

The aim of this trial is to assess the effects of group interpersonal synchrony on the prosociality and work-related stress of young autistic adults in their work environment. The trial is designed as a randomized controlled trial with two parallel groups (experimental group and treated control group), primary and secondary endpoints and a follow-up of intervention effectiveness after 17 weeks as described in Fig 1. Randomization will be performed as block randomization with a 1:1 allocation. In the following sections we review research on interpersonal synchrony in autism and on the associations between interpersonal synchrony, prosociality and work-related stress.

### Interpersonal synchrony in autism

Interpersonal synchrony is said to occur when two or more individuals are engaged in body movements or sensations (such as gaze, affect, voice and touch) at the same time (precise synchrony) or as a responsive behavior (lagged synchrony) [17, 18]. This form of interpersonal relationship has at times been called "togetherness", especially when there is no leader or

| | STUDY PERIOD | | | | | |
| --- | --- | --- | --- | --- | --- | --- |
| | Enrolment | Random-ization | Post-allocation | | | |
| TIMEPOINT | 1st week of the course | 0 | 2nd week of the course | 13th (last) week of the course | two weeks after end of course | 17 weeks after end of course |
| **ENROLMENT:** | | | | | | |
| *Eligibility screen* | X | | | | | |
| *Informed consent* | X | | | | | |
| *Allocation* | | X | | | | |
| **INTERVENTIONS:** | | | | | | |
| *Synch Group Intervention* | | | ●————————————● | | | |
| *Non-Synch Group Intervention* | | | ●——————————● | | | |
| **ASSESSMENTS:** | | | | | | |
| *Demographics; Medical; Physiological; EHQ* | X | | | | | |
| *NTB* | X | | | X | | X |
| *IOS; FCI; GSB; IS; CT; PGG* | | | X | X | | X |
| *Semi-structured interview* | | | | | X | |

**Fig 1. Schedule of enrolment, interventions and assessments.** Note: EHQ, Edinburgh Handedness Questionnaire; NTB, Need to Belong Scale; IOS, Inclusion of Other in Self Scale; FCI, Friendship Closeness Inventory; GSB, General Sense of Belonging Scale; IS, Irritation Scale; CT, Collection Task; PGG, Public Goods Game.

follower in this form of non-verbal behavioral sharing [19]. Body movements may be connected spatially (spatial synchrony), temporally (rhythmic synchrony) or qualitatively (effort synchrony) [20, 21]. Interpersonal synchrony can emerge intentionally as a result of an explicit shared goal, such as running at same pace, or spontaneously without being aware of it as part of a social interaction, such as when dancing, singing or playing an instrument [22]. Studies over the last ten years have shown that autistic children and adults demonstrate poor ability to synchronize their body with a partner during rhythmic interaction activities and games [11, 12, 23–26]. For example, an innovative study focusing on motion synchronization during an open-ended joint improvisation paradigm dubbed the Mirror Game found that cognitively-abled autistic adults exhibited much less interpersonal synchronization than neurotypical individuals [23]. Autistic adolescents also presented less interpersonal synchrony, and fewer intentional and spontaneous motor interactions on a pendulum coordination paradigm [12].

## The effects of interpersonal synchrony on prosociality

Prosociality refers to behaviors that are intended to benefit others, such as helping, cooperating, sharing and donating [27]. Recently, there has been increased interest in studying the causal influences of interpersonal synchrony on prosociality in general [28, 29] and specifically in work environments [30, 31]. A meta-analysis of 60 studies that compared an interpersonal synchrony condition to at least one control condition found a medium-sized effect of interpersonal synchrony on the prosociality of neurotypical adults with regard to both attitudes and behavior towards individuals or groups involved in the same intervention [28]. Another meta-analysis of 42 studies suggested that synchrony in larger groups increases prosociality [29]. Other studies that have assessed the effects of interventions based on mirroring movement in autistic adults suggest that interpersonal synchrony may be an effective intervention tool for enhancing social skills [32] and specifically emotional inference [33], which is a fundamental skill underlying prosociality. Manders et al. [34] argued that the structured mirroring task, which encourages interpersonal synchrony such as in the case of the Mirror Game, may not be sufficient to effectively support social engagement in autistic individuals, since they tend to simply follow the instructions to synchronize their movements. These findings emphasize the need to include synchronized sensory-motor interactions in tailored interventions to foster prosociality, while proposing both structured intentional synchrony and less structured spontaneous synchrony [35]. To the best of our knowledge, no study has examined the effectiveness of group interpersonal synchrony on the prosociality of autistic individuals in the context of a social work environment. Since working teams' cognitive performance declines in stressful environments [36], in the following section we review research on the associations between interpersonal synchrony and work-related stress.

## The effects of interpersonal synchrony on work-related stress

Work-related stress is defined as individuals' responses when presented with work demands and pressures that are misaligned with their knowledge or abilities and which challenge their ability to cope [37]. A recent longitudinal field experiment in a neurotypical work environment found that a 9-week synchronous movement intervention reduced work-related stress and diminished sick days immediately following the intervention, but not after three months [30]. A qualitative evaluation of workplace choir singing in Health Service staff reported greater enjoyment at work and an improvement in work engagement [31]. One possible explanation is that participating in synchronous activities may release endogenous opioids such as endorphin, which are involved in the regulation of emotional pain [38]. While these studies support the widespread belief that togetherness and harmony only have beneficial effects on personal well-being, a study by Galbusera et al.'s [39] suggested that interpersonal synchrony when implemented in the form of a dyadic moving task designed to encourage spontaneous social interaction, could decrease people's ability to self-regulate due to their extra reliance on the dyadic interaction. This may imply that an effective intervention to support autistic individuals in stressful work environments should include co-regulation with others in a group setting rather than in an intimate dyadic framework that might be less efficient in terms of stress regulation. To the best of our knowledge, no studies have examined the effectiveness of interpersonal synchrony on the work-related stress of autistic individuals in the context of a group.

## The effects of interpersonal synchrony on social closeness and a sense of belonging

According to McNeill's [40] Muscular Bonding hypothesis, group interpersonal synchrony, which involves coordinated rhythmic movement, is a powerful force uniting human groups

and enhancing social closeness and a sense of belonging. Social closeness is defined as the way individuals engage in friendships, interact and relate to significant others [41]. Social closeness in a workplace setting was enhanced using an intervention in which participants shared synchronous movement [30]. Sense of belonging is the experience of personal involvement in a system or environment, which makes individuals feel that they are an integral part [42]. Interpersonal synchrony has been found to enhance people's sense of belonging in an experimental setting using a finger tapping task [43]. Furthermore, these feelings of social closeness and belongingness among individuals are known to promote prosocial behavior [44] as well as the ability to cope with various stressors [45].

## Moderating factor underlying the effects of interpersonal synchrony

The need to belong is characterized by the drive to be noticed, valued and accepted by others [46]. It is a fundamental need that exerts a strong influence in virtually every domain of social behavior starting early in development [47]. While all individuals differ in the strength of their desire to belong to social groups, the social motivation hypothesis argues that autistic individuals' need to belong is diminished because they find social stimuli less rewarding as compared to neurotypicals [4]. For example, Brezis et al. [23] found that autistic adults who played the Mirror Game with an expert improvisor were less likely to want to continue playing, but their ratings of how they felt during the game did not differ. Hence, group interpersonal synchrony may be more effective when the need to belong is higher, since the intrinsic motivation is likely to have a positive effect on the sense of belonging to the group. The suggested moderating model is depicted in Fig 2.

## Objectives

The aim of this study is to assess the effects of group interpersonal synchrony on the prosociality and work-related stress of young autistic adults in their work environment. The primary objective is to determine whether a synchronized group intervention will have an immediate and/or long-term positive effect on participants' prosociality and work-related stress. The secondary objectives are to determine: (1) Whether a synchronized group intervention will have an immediate and/or long-term positive effect on participants' social closeness and sense of

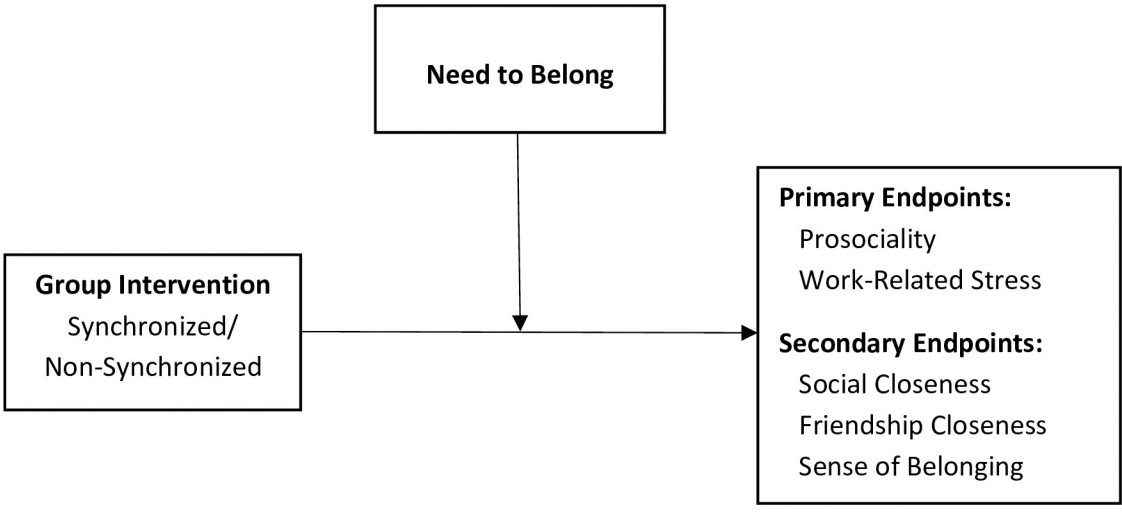

**Fig 2. Theoretical moderating model of interpersonal synchrony.**

belonging, (2) Whether this effect will be influenced by participants' need to belong as reported before the intervention, (3) How participants perceive the intervention as affecting their prosociality and work-related stress, and (4) In what ways the participants' perception will contribute to a better understanding of the intervention effect.

## Materials and methods

A convergent mixed methods design will be applied, where quantitative and qualitative data are collected and analyzed in parallel. The use of mixed methods may provide a better understanding of the study objectives. Specifically, a convergent design will be used [48] to gain a more in-depth understanding of the quantitative results by incorporating the participants' qualitative perspectives. The two sets of results will be compared to obtain a more complete understanding of the study objectives by validating one set of findings with the other and ensuring that the participants' perspectives are consistent and not dependent on the research method.

### Participants

This study will be conducted in collaboration with *Roim Rachok* Program (RRP) an innovative Israeli civilian program designed to integrate cognitively-abled young autistic adults first into the Israeli army work force, and later into the labor market [6]. As part of the program, before joining the army, the trainees take part in a professional 13 weeks course delivered by an interdisciplinary team that includes army officers and instructors, an occupational therapist, a speech therapist and an art therapist. The course content is composed of two main fields: (a) army related professional instruction (e.g., software programming, data analysis and image interpretation), and (b) integration into the army working environment. Trainees who successfully complete the course are inducted into the army as civilian volunteers for a four-month trial period. The trial period is designed to allow trainees to experience their future working environment before committing to long-term army service. Trainees who successfully complete the trial period can sign up for one more year of service with an annual option of extension, for up to three years. The RRP participants are integrated into the army corps with other soldiers who do not have developmental disabilities.

The sample for this study will be comprised of young adults enrolled in an RRP who fulfill the following criteria: **Inclusion criteria:** (a) Aged 18 to 25 when accepted to the program, since these age limits are defined as part of the admission requirements for the RRP. (b) Diagnosis of autism spectrum disorder: Participants must have an official diagnosis of an autism spectrum disorder as assessed by a child psychiatrist or clinical psychologist according to the DSM-V [1] before acceptance to the RRP. **Exclusion criteria:** Trainees with severe sensory impairments such as blindness or deafness and/or severe physical disability.

### Ethical considerations

This study has been approved by the ethics committee for human experiments of the Faculty of Social Welfare and Health Sciences at the University of Haifa (Approval # 017/21). Participation in the study will be voluntary without preconditions. All participants will receive a letter explaining the study and how it will be conducted, and will sign an informed consent form to participate in the study. Participants who have custodian will be required to present a written custodian approval as well. No risks or discomfort are expected as a result of participation in this study. However, if any physical or emotional discomfort is experienced, the participants will receive appropriate support from the RRP professional staff. The data will be coded to protect the participants' confidentiality. All data, including the questionnaires, the behavioral task

outcomes and audio recordings will be stored in a secure cloud and will not be available to anyone other than the research team. The audio recordings will be deleted immediately after being transcribed.

## Sample size

A meta-analysis on prosocial consequences of interpersonal synchrony among neurotypical adults found a medium-sized effect of interpersonal synchrony on prosociality with regard to both attitudes and behavior towards individuals or groups involved in the same intervention [28]. Therefore, an effect defined as d = 0.25 may be expected [49]. An a-priori power analysis using the G*Power computer program [50] indicated that a total sample size of 40 participants would be needed to detect medium-sized effects defined as f = 0.25, with 80% power and alpha at .05, using a repeated measure, within-between interaction ANOVA with four groups and three measurements. We will recruit at least N = 60 participants (30 in each intervention group) to plan for possible dropouts.

## Participant recruitment

A total of 60 participants will be recruited at the beginning of a program cycle from approximately four consecutive program cycles, each of which is composed of 20 to 30 trainees. A new program cycle starts every four months. The first author will be responsible for the study approval procedure at the beginning of the course while providing information about the study to all trainees and inviting them to participate. The trainees' decision to participate or not in the study will not affect their participation in the course in any way. The trainees who agree to take part in the study will sign an informed consent form that clarifies that participation in the study is on a voluntary basis and that anonymity and confidentiality are guaranteed as well as the right to withdraw at any time without this affecting their participation in the program. Participants who have a custodian will be required to present the custodian's approval as well. This form will also include information about the data collection procedures, including data from the RRP personal file.

## Study design

This study will implement a randomized controlled trial (RCT) design to investigate a synchronous or non-synchronous movement-based group intervention, so that each participant will only engage in one intervention condition, alongside a phenomenological study with the participants of both groups. The RCT will compare the experimental group (synchronous movement) with a treated control group (non-synchronous movement) and will follow the recommendations of the Consolidate Standards of Reporting Trials (CONSORT) [51] and the recommendations of the Standards for Reporting Qualitative Research [52]. In addition, the SPIRIT guidelines [53] have been followed. Fig 1 outlines when each study component occurs. This design does not include a nontreated control group to meet the program's requirement that all trainees should participate in all course activities including the movement-based group intervention. The study was registered with ClinicalTrials.gov (registration number: NCT05846308) afterthe enrolment of participants started due to the uncertainty about the clinical character of the target population and intervention. The authors confirm that all ongoing and related trials for this intervention are registered.

The condition groups will be matched according to the following control variables: (a) gender (male/female); since there is diagnostic gender bias with respect to ASD [54] (b) army related profession (image interpretation/GIS/data analytics/software programming/electro-optics/data annotation) (c) handedness (right-handed/left-handed); since there is a larger

distribution of left-handedness in the ASD population compared to the general population [55] (d) need to belong level (NTBL) (high/low). To control for handedness and NTBL during the randomization process, the following questionnaires will be administered to the participants: (a) The Hebrew adaptation of the Edinburgh Handedness questionnaire [56], the most common method for measuring handedness in individuals with Autism [57]. (b) The Hebrew adaptation of the Need to Belong (NTB) Scale [59] which comprises ten items rated on a 5-point Likert scale ranging from "Strongly Disagree" (1) to "Strongly Agree" (5). This scale has acceptable interitem reliability (the Cronbach's alpha generally exceeds .80) and has been positively correlated with extraversion, agreeableness, and neuroticism and with having an identity that is defined in terms of social attributes [58]. NTBL(high/low) will be calculated using a cut-off of the NTB measure, so that participants with the 50% highest NTB score will be assigned a high NTBL and the rest will be assigned a low NTBL. After inclusion in the study and assessment of the control variables, participants will be assigned by the first author to one of the intervention conditions on an individual basis according to a computer-generated randomization procedure. The allocation ratio of the intended numbers of participants in the comparison groups will be 1:1 so that the number of participants assigned to the synchronous intervention will be similar to the number of participants assigned to the non-synchronous intervention. To this end, we will first create subgroups containing participants with same gender, army-related profession, handedness and NTBL. For each subgroup, alternate assignments will be made in random order. The order of subgroup assignments will also be determined randomly. For the first subgroup, we will use a simple coin flip randomization procedure to determine the intervention type to start assigning to. An overview of the study design is shown in Fig 3.

## Procedure

The interventions will be conducted by the first author, who is a professional Dance Movement therapist, as part of the RRP course, once a week for the duration of the 10 weeks of each program cycle. A graduate student of the RRP will take part as a co-instructor for modelling and assistance. Each intervention group will have between 10 and 14 participants. Since autistic participants may feel anxious without a predictable structure, a structured physical training protocol will be used for each condition. The protocols were designed by the first author with the help of a professional physical trainer and after consulting with dance movement therapists who have research experience in the area of synchrony interventions in ASD. Each protocol is composed of 10 physical training sessions, each lasting 60 minutes. The protocols differ in terms of using synchronous activity vs. non-synchronous activity. They do not differ in terms of physical exercise type or duration to control for the effect of exercise type and duration on the dependent variables. Some exercises include the use of a training mattress, an elastic band or a sponge ball. To allow for some flexibility in terms of spontaneous synchrony, each session includes a social engagement segment either as a warm-up or as a social game. A Tabata Timer [59] will be used to break the workout into clearly defined intervals during the main training segment. A metronome beat (every second) will be used to help the participants do the exercises at the right pace.

## Synchronous condition

The instructors and the participants will form a circle facing each other while doing the physical exercises as shown in Fig 4. To facilitate interpersonal synchrony, the participants will be instructed to do the same physical exercises (spatial synchrony) together at the same pace (rhythmic synchrony). The participants will not be instructed to use the same amount of effort, to accommodate a range of strengths and intensities.

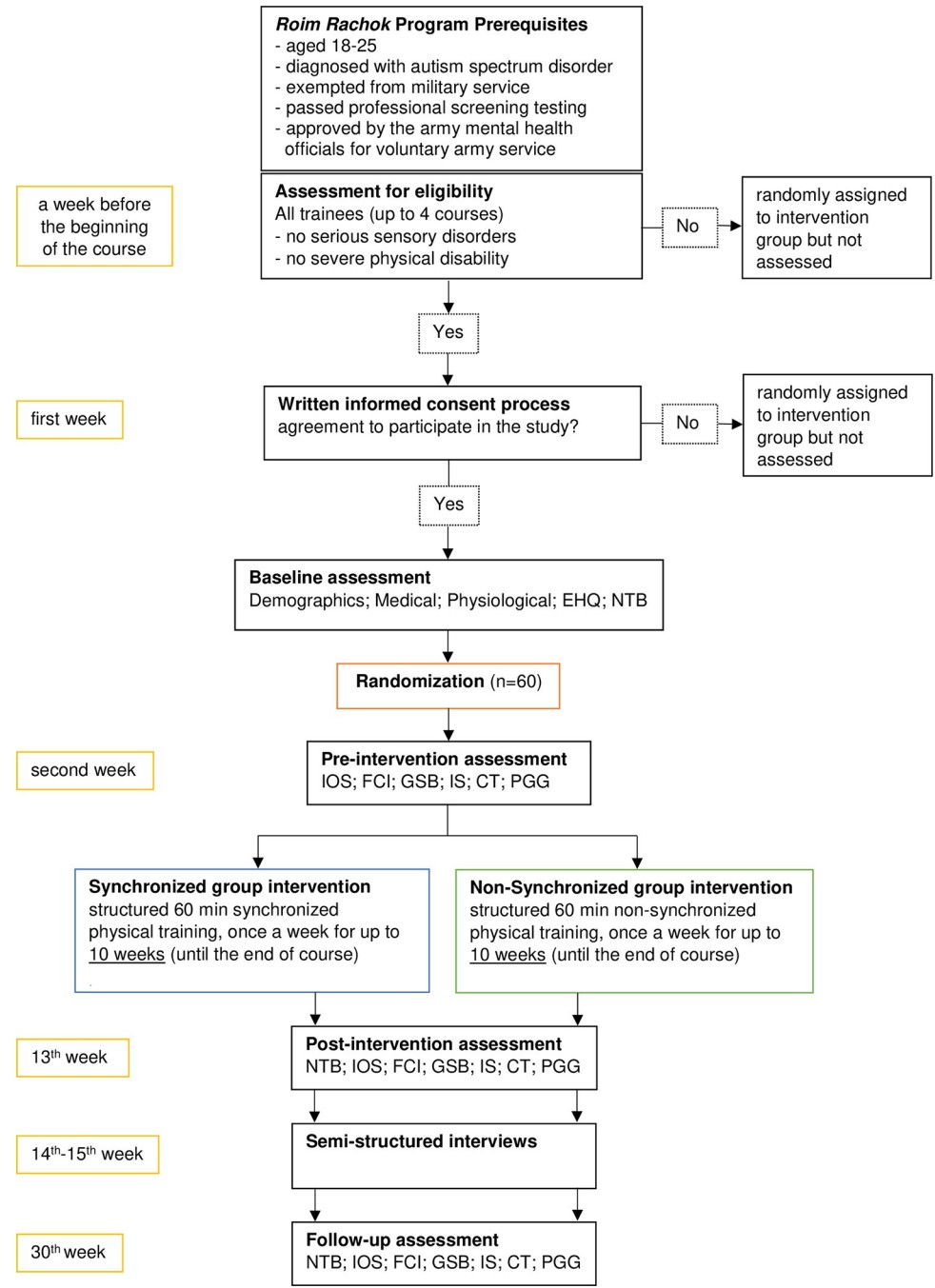

**Fig 3. Flow of participants through the study: Illustration of the study design.** Note: EHQ, Edinburgh Handedness Questionnaire; NTB, Need to Belong Scale; IOS, Inclusion of Other in Self Scale; FCI, Friendship Closeness Inventory; GSB, General Sense of Belonging Scale; IS, Irritation Scale; CT, Collection Task; PGG, Public Goods Game.

## Non-synchronous condition

The participants will do the same physical exercises as the participants in the synchronous group but in the form of circuit training with seven stations. The circuit training will require the participants to do a different physical exercise at a different pace at each station. A detailed description of the exercises will be provided for each station. The instructors will demonstrate all the

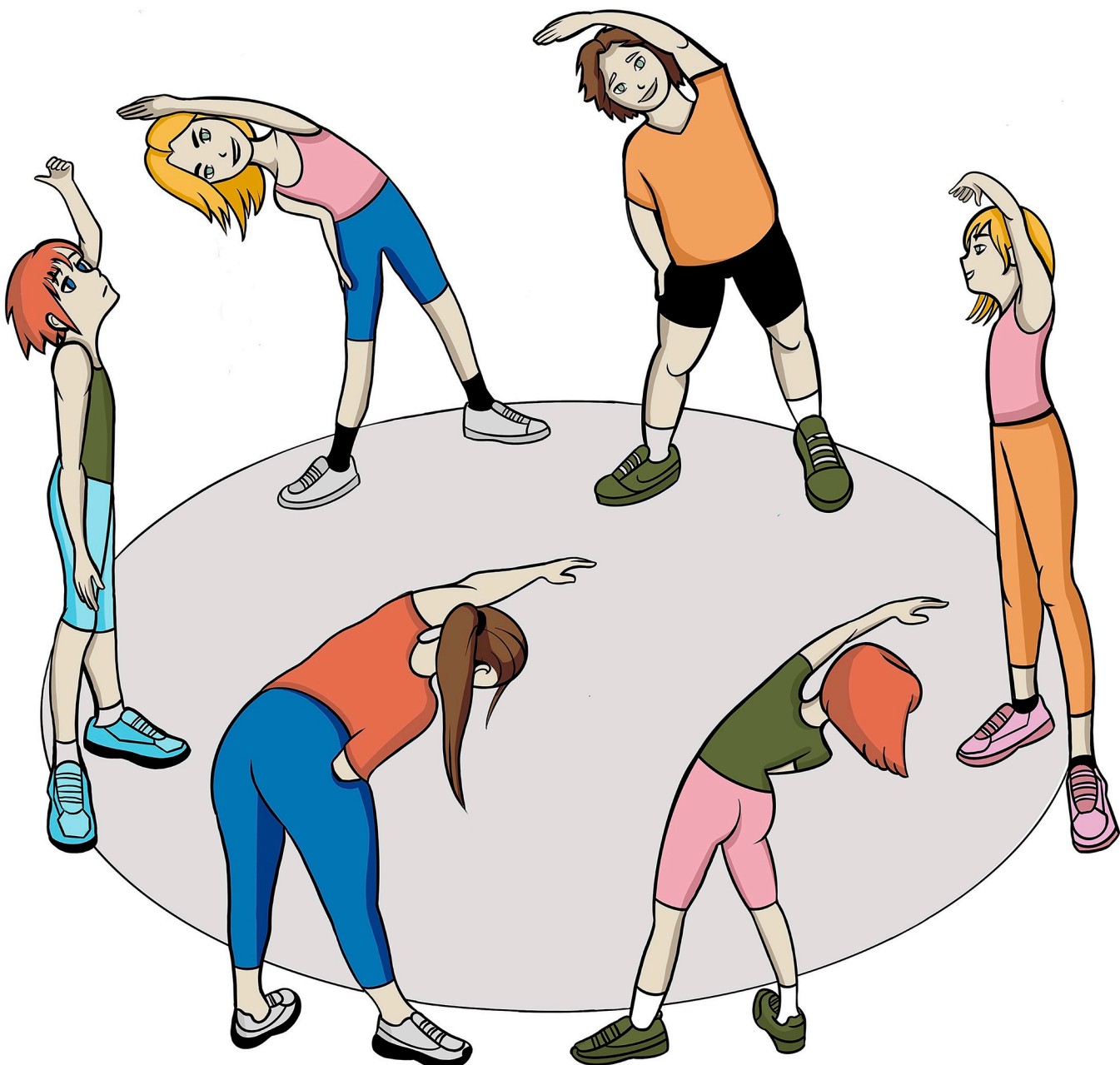

**Fig 4. Synchronized group intervention.**

exercises before the beginning of training. The participants will be instructed to do the exercise for a set period of time, the same duration used in the synchronous group. The circuit training stations will be in the form of a circle, as shown in Fig 5, but will be set up so that the participants do not to face each other when doing the exercises to prevent spontaneous synchronization. Depending on group size, the participants will work either as dyads or as single trainees in each station. In the dyad form one participant will do the exercise while the partner is resting and vice-versa. In the single form each participant will exercise as though a partner were present. The participants will be instructed to change partners each week so that each participant will have the opportunity to experience both the single and dyadic forms, with different partners.

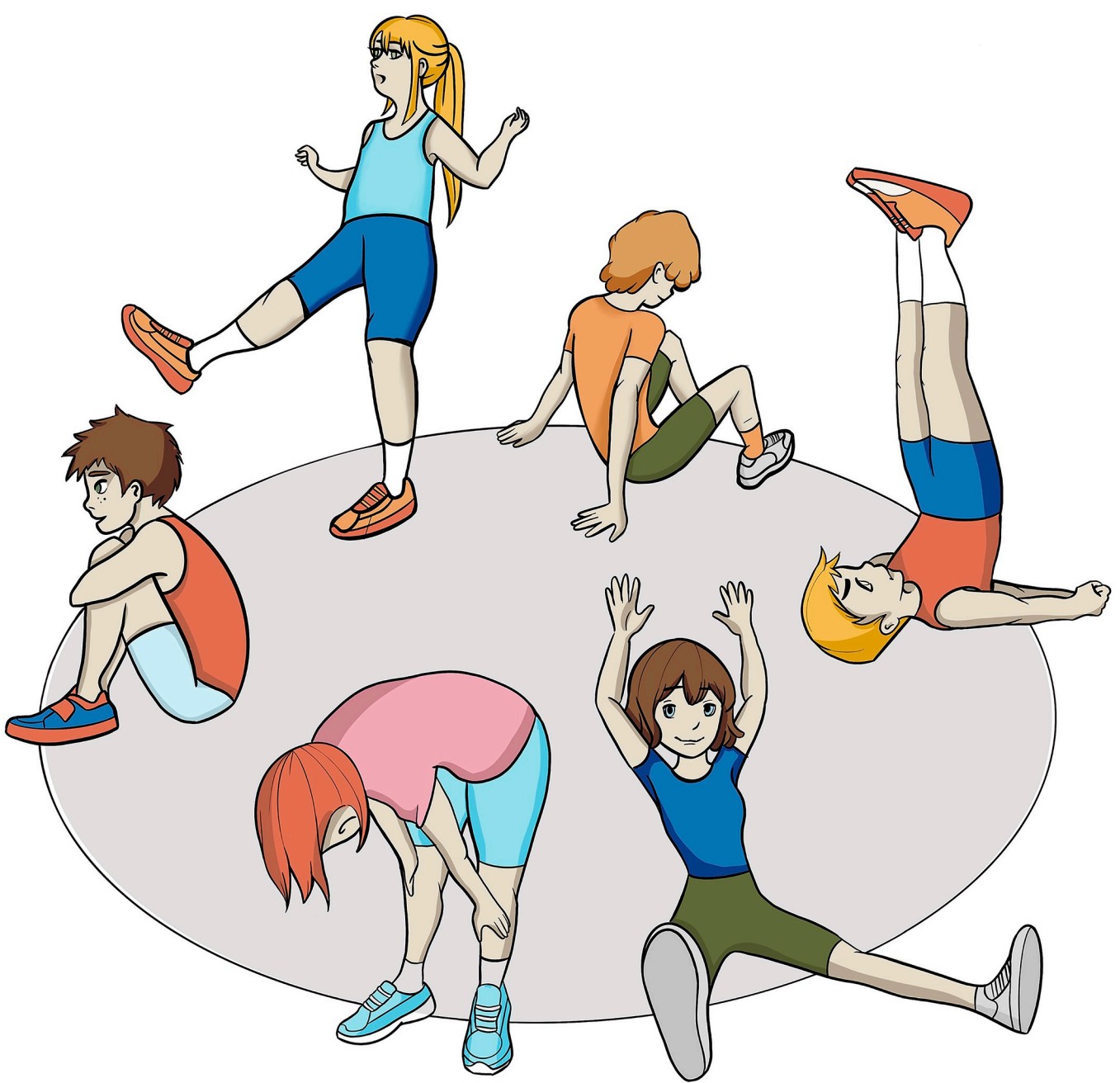

**Fig 5. Non-synchronized group intervention.**

## Intervention protocol

Each session will consist of four parts as listed in Table 1.

## Quantitative data collection: Background and control variables

Background and control data will be collected for each participant from the RRP personal file which contains demographic and medical information obtained by the program staff using dedicated questionnaires. The questionnaires are handed out routinely to all RRP trainees

**Table 1. Intervention protocol.**

| Part | Sub part | Synchronous condition | Non-synchronous condition |
|---|---|---|---|
| **Warm-up** (10 minutes) | Body joints warm-up: neck, shoulder, pelvis, hip and ankle | Rotating the joints four times on each side of the body together at the same pace. | Rotating the joints four times on each side of the body, at each participant's own pace. |
| | Cardio warm-up: running, jumping skipping | Moving together in a circle while instructed to keep the same pace. | Moving in a form of relay race, at each participant's own pace. |
| **Social Game** (10 minutes) | Each session will contain a different movement game such as catch and "numbers in a circle" (outdoor training game). The social flexibility of the game will be gradually increased during the intervention period thus allowing for spontaneous synchrony. | | |
| | | For example: in the "catch" game the participants will have to pass a ball to each other 21 times simultaneously, thus depending on each other to successfully complete the game. | For example: in the "catch" game the participants will have to bounce the ball against a wall 21 times and catch it, thus depending on their own performance to successfully complete the game. |
| **Main training** (30 minutes) | Each session will contain two sets, each consisting of seven fixed exercises with a resting period of 10–15 seconds between each. Each exercise combines strength only (20–30 sec) or strength (20–30 sec) and cardio (15 sec). There will be a resting period of 3 minutes between each set of exercises. Each exercise will be performed at one of three different paces: *Slow*—one movement cycle per 2 seconds, *Medium*—one movement cycle per 1 second and *Fast*—two movement cycles per 1 second, to allow for the non-synchronous group to do the same exercises as the synchronous group in non-synchronized way (both spatial and rhythmic). | | |
| | Legs and upper body set (e.g., squats, heel raising, arm rotations etc.) | Performing the same physical exercises together at the same pace in a circle facing each other. | Performing different physical exercises at a different pace in a circuit training setup. |
| | Back and abdomen set (e.g., pushups, planks, crunches etc.) | | |
| **Cool-down** (10 minutes) | Muscle stretching: arms, hands, thighs, legs and feet | Stretching eight counts on each side of the body together at the same pace. | Stretching eight counts on each side of the body, at each participant's own pace. |
| | Relaxing | Performing eight breathing cycles together at the same pace. | Performing eight breathing cycles, at participant's own pace. |

during the program selection process, a few months before admission to the program. These cover information such as demographic data (e.g. age, gender, country of birth, residential area, number of siblings and birth order), medical data (e.g. diagnosis, age at diagnosis, comorbidities, use of psychiatric medication and HMO disability bracket) and physiological data (e.g. body weight and height).

## Quantitative data collection: Outcome variables

Quantitative data on the following outcome variables (as listed in Table 2) will be collected for each participant using validated questionnaires and behavioral tasks at three points of time: before the first intervention session, after the last intervention session (pre-post) and 17 weeks after the end of the intervention period (follow-up). The data collection procedure will be

**Table 2. Study outcomes.**

| Outcomes |
|---|
| **Primary Outcomes** |
| Work-related stress |
| Non-confidential behavioral cooperation |
| Confidential behavioral cooperation |
| **Secondary Outcomes** |
| Social closeness |
| Friendship closeness |
| Sense of belonging |

administered by the first author in a group session, corresponding to the same group as the intervention group, in a classroom at RRP facilities. This procedure will be composed of two parts: (a) A group behavioral task which will be recorded using one video camera. (b) Questionnaire administration via online software (Qualtrics).

**Social closeness.** Social closeness will be measured using the Inclusion of Other in Self Scale (IOS) [60, 61]. This scale is made up of seven Venn diagram-like pictures where one circle represents the participant and the other circle represents the entire intervention group. The diagrams, ranging from no overlap to near complete overlap, measure close relationships between self and the group by capturing aspects of both feeling close and behaving close. The scale has high reliability for the friendship subgroup sample (Cronbach's alpha .92) and a high test-retest correlation (r = .86 (n = 31)). Significant correlations between the IOS and measures of closeness that are primarily verbal and multi-item were found (60).

**Friendship closeness.** Friendship closeness will be measured using the Hebrew adaptation of the Friendship Closeness Inventory [41]. The FCI is composed of 49 items that measure closeness in same-sex friendships and is divided into three distinguishable yet related subscales: Emotional Closeness (EC), Behavioral Closeness (BC), and Cognitive Closeness (CC). For the purposes of the present study the first item will be adjusted to include a reference to participants in the intervention group alone: "*Do you have friends among the participants in the physical training group whom you consider to be 'close friends'*?". The scale has high reliability for both the total score (Cronbach's alpha .94) and for the subscales scores (Cronbach's alpha .91 for EC, Cronbach's alpha .93 for BC and Cronbach's alpha .87 for CC) [41].

**Sense of belonging.** Sense of belonging will be measured using the Hebrew adaptation of the General Sense of Belonging Scale [62]. This scale is composed of 12 items that measure sense of belonging (achieved belongingness) which are rated on a 7-point Likert scale ranging from "Strongly Disagree" (1) to "Strongly Agree" (7). Six items measure sense of acceptance and inclusion; for example: *I have a sense of belonging*. The other six items measure sense of rejection and exclusion; for example: *I feel like an outsider*. For the purposes of the present study, the scale will be adjusted so that the words "other people" or "others" will be replaced by the words "participants in the physical training group"; for example: "*When I am with participants in the physical training group, I feel included*". The scale has high reliability (Coefficient alpha = .94 and an average inter-item correlation (AIC) = .57 (M = 66.3, SD = 13.5) and strong patterns of validity estimates have been established (62). The original version of the General Sense of Belonging Scale [62] will also be used to validate the adapted one.

**Work-related stress.** Work-related stress will be measured using the Hebrew adaptation of the Irritation Scale [63]. This scale comprises eight items, three of which assess cognitive irritation and five of which assess emotional irritation. Items are rated on a 7-point Likert scale ranging from "Strongly Disagree" (1) to "Strongly Agree" (7). For the purposes of the present study, the scale will be slightly adjusted in that the words "at work" will be replaced by the words "in the course", for example: "*Even at home I often think of my problems in the course*". The scale has high reliability (Cronbach's alpha .89) and for the German version, a large number of studies report correlations between this scale and stressors at work, further impairments such as psychosomatic complaints or depression, and missing resources such as social support [63].

**Prosociality.** Prosociality will be measured using the following behavioral group cooperation tasks, both of which resemble the classic N-person cooperation dilemma as a "collective action problem of a common good" as described by [64].

*1. Collection task.* Participants will have to work together to pick up 100 small washers (a flat plastic coin with a diameter of 4 cm), as described in [65]. The participants will be asked to remain quietly in their seats for 5 minutes while the washers are being randomly scattered in a

rectangular structure on one side of the classroom. Then the participants will be asked to pick up one washer at a time and put it in a small basket located on the other side of the classroom in the shortest time, until all washers are collected by the group. Only then will they be able to return to their seats and remain quietly for another 5 minutes. Participant will be asked to abstain from running to avoid collisions and maintain group safety. In this specific task, the act of gathering washers embodies the concept of the common good. Participants must collect all washers and place them in a small basket before returning to their seats. However, the temptation to free ride, relying on others' efforts, illustrates the collective action problem. While it benefits both individuals and the group to gather the washers, the impact of any single person's contribution is minimal and difficult to discern. This dynamic creates an incentive to conserve energy by abstaining from action and allowing others to undertake the task. Participants must work together toward a common goal (collecting all washers in the shortest time), therefore there is likely no incentive to compete. As in previous social loafing paradigms [66], cooperation on this task will be operationalized by participants' effort to overcome the incentive to conserve energy, represented by their step rate (number of steps per second). The number of steps will be measured using a wearable fitness tracker (Fitbit Inspire 2) that will be attached to the participants' wrists using a special band. Fitbit fitness trackers are commonly used in clinical trials [67]. Each tracker will be customized with participants' personal height and weight to ensure step rate accuracy. The collection procedure will be recorded using one video camera to control task execution. Since cooperation on this task will depend on the participants' physical fitness and might be influenced by group pressure, we will also use a non-physical confidential task to measure pure voluntary cooperation.

*2. Public goods game.* Participants will be told they will be given 30 NIS and that they can donate some or all of it to a group investment, as described in [68]. The money in the group investment will then be doubled and divided equally among all members of the group. The money earned in this game will be compensation for participating in the study. In contrast to the collection task, participants' contribution will be confidential to avoid group pressure. Cooperation on this task will be operationalized by the amount of money each participant decides to donate to the group investment.

## Qualitative data collection

Qualitative data will be collected from the participants till data saturation is achieved, using semi-structured interviews. After each intervention cycle, all participants will be asked to volunteer to be interviewed. The interviewees will be chosen out of the pool of volunteers based on the following criteria: (a) At least one participant from each intervention group. (b) At least one participant from each army related profession. (c) High verbal self-expression—the participant must have been appraised by army officers as being able to verbally express thoughts, emotions and views. The interviews will be carried out one to two weeks after the end of the intervention cycle. The interview guide for the semi-structured interviews will be composed of 11 questions ranging from the most general (e.g., "*How did you experience the course in general*?") to the most specific (e.g., "*How did the group physical training sessions affect your experience during the course: (a) with respect to your sense of belonging (b) with respect to friendship closeness (c) with respect to your sense of involvement and cooperation (d) with respect to your work-related stress*"). In order to merge the qualitative and quantitative data, the qualitative questions in the interview guide will be congruent with the quantitative objectives of this study. To motivate the participants to discuss their perspective and express their emotions, two questions out of the 11 will be administered using a set of projective cards. This set of cards contains 13 words and 17 pictures reflecting associations with the word "group". On the

first question the participants will be asked to choose a word that best describes their relationship with other members of the intervention group (e.g. "coherence", "loneliness" and "opportunity") and to explain why they chose this specific word. On the second question the participants will be asked to choose the picture that best depicts their experience during the group intervention process and to explain why they chose this specific picture. The use of semi-ambiguous words and pictures is less challenging for individuals on the autistic spectrum than the presentation of more ambiguous stimuli. The interviews will be conducted by the first author and will be audio recorded and fully transcribed before analysis.

## Data analyses

To answer the five study objectives (see above in Introduction) the following data analysis will be conducted:

**Quantitative data analysis.**   The data obtained as study variables will be coded, processed and analyzed at the end of the collection process. IBM SPSS Statistics Version 27 will be used for statistical analysis. To control the family-wise error rate when performing multiple hypothesis tests, Bonferroni Step-Down (Holm) procedure will be applied. Participants must attend at least 4 sessions to be included in the analyses. To describe the sample, descriptive statistics will be calculated followed by univariate and bivariate statistical analyses. The first step will be to assess whether the data follows a normal distribution for each of the variables which will guide the choice of parametric or non-parametric tests. To answer the primary study objective and the first secondary study objective, the effects of the intervention over time (at the end of the intervention and after 17 weeks) will be examined using a two-way mixed model ANOVA. To explore the second secondary study objective, the influence of the need to belong on the effect of the intervention over time (at the end of the intervention and after 17 weeks) will be examined using a three-way mixed model ANOVA.

**Qualitative data analysis.**   To investigate the participants' experiences as related to the third secondary study objective, the semi-structured interviews will be analyzed using a thematic analysis approach [69, 70] in which the data are systematically parsed to identify patterns of meanings (themes). The analysis will be performed using MAXQDA 2022.

**Mixed methods analysis.**   For data integration related to the fourth secondary study objective, a merged data analysis will be conducted based on side-by-side comparison of the quantitative findings with the interview data [48].

## Study status

This study is currently ongoing. A feasibility assessment [71] of the intervention protocol was conducted to fine-tune the intervention protocol by documenting the experiences of the participants in terms of the goals of each of the intervention conditions (see above in Procedure). Participants' allocation to intervention conditions during the feasibility assessment was not randomized or controlled and no data was collected. Full RCT implementation during three program cycles (starting in August 2021) have been completed successfully. The RCT implementation of the fourth program cycle is in process.

## Discussion

Serving in the army in Israel is mandatory at the age of 18 and represents the first step towards independent life outside of the home. Until recently, cognitively-abled autistic adults were automatically exempt from military service. The *Roim Rachok* civilian program aims to develop professional skills and tools to enable the successful integration of this population into an army work environment [6]. Recently, a new army program was developed to enable a full

mandatory service for cognitively-abled autistic adults on a larger scale, and is currently being pilot tested. Clearly more needs to be known about interventions that promote successful integration in social and stressful work environments.

This study constitutes an innovative effort to empirically examine the use of group interpersonal synchrony as an effective intervention for young autistic adults who would like to successfully integrate into the Israeli army work force. It can contribute to the enhancement of the state of the art in the area of interpersonal synchrony as an effective intervention for autistic individuals specifically, and for individuals with difficulties in adapting to social and stressful work environments in general. The findings are also expected to contribute to the evidence base on intervention options for young autistic adults. The results are likely to provide information on the relevance of synchronized intervention as compared to non-synchronized interventions for individual wellbeing by incorporating the participants' experiences. The findings can also contribute to further specify treatment guidelines for this population and to enrich future training and education of dance movement therapists, music therapists and other health care professionals working in the field of ASD.

The strength of this study lies in its ability to conduct an RCT in a field setting where the interventions are embedded in a plausible context: a program that aims to improve participants' work-related soft skills. The trainees' motivation to participate in this study is expected to be high, since both the intervention and data collection processes are embedded within the program schedule with no need for extra time-consuming activities (except for the individual interviews). The strength of this study is also linked to its potential limitations in terms of quantitative data generalizability due to the relatively medium sample size and the heterogeneity of the population. We expect that merging the quantitative results with in-depth qualitative outcomes will facilitate the transferability of the findings to other contexts.

## Supporting information

**S1 Checklist. SPIRIT 2013 checklist: Recommended items to address in a clinical trial protocol and related documents\*.**
(DOC)

**S1 File.**
(DOCX)

## Acknowledgments

We thank the following individuals for their voluntary collaboration and support of this study: Tal Vardi (RRP Manager), Dr. Efrat Selanikyo (RRP Professional manager), Eyal Ron (RRP Course manager), Ori Dvir (computer science student) for helping develop the fitness tracker data retrieval software, Tomer Wasserman (certified physical trainer) for helping in the design of the intervention protocol, and Elad Shtayinmatz (RRP graduate) for helping administer the intervention protocol.

## Author Contributions

**Conceptualization:** Tamar Dvir, Tal-Chen Rabinowitch, Cochavit Elefant.

**Data curation:** Tamar Dvir.

**Investigation:** Tamar Dvir.

**Methodology:** Tamar Dvir, Tal-Chen Rabinowitch, Cochavit Elefant.

**Project administration:** Tamar Dvir.

**Resources:** Tamar Dvir.

**Software:** Tamar Dvir.

**Supervision:** Tamar Dvir, Tal-Chen Rabinowitch, Cochavit Elefant.

**Visualization:** Tamar Dvir.

**Writing – original draft:** Tamar Dvir.

**Writing – review & editing:** Tamar Dvir, Tal-Chen Rabinowitch, Cochavit Elefant.

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
