## [Decision Letter · Decision Letter 0]

7 Feb 2024

PONE-D-23-06520The effectiveness of group interpersonal synchrony in young autistic adults’ work environment: A mixed methods RCT study protocolPLOS ONE

Dear Dr. Dvir,

Thank you for submitting your manuscript to PLOS ONE. After careful consideration, we feel that it has merit but does not fully meet PLOS ONE’s publication criteria as it currently stands. Therefore, we invite you to submit a revised version of the manuscript that addresses the points raised during the review process.

We look forward to receiving your revised manuscript.

Kind regards,

Stergios Makris

Academic Editor

PLOS ONE

Journal Requirements:

Reviewers' comments:

Reviewer's Responses to Questions

**Comments to the Author**

1. Does the manuscript provide a valid rationale for the proposed study, with clearly identified and justified research questions?

Reviewer #1: Yes

2. Is the protocol technically sound and planned in a manner that will lead to a meaningful outcome and allow testing the stated hypotheses?

Reviewer #1: Partly

3. Is the methodology feasible and described in sufficient detail to allow the work to be replicable?

Reviewer #1: Yes

4. Have the authors described where all data underlying the findings will be made available when the study is complete?

Reviewer #1: No

5. Is the manuscript presented in an intelligible fashion and written in standard English?

Reviewer #1: Yes

6. Review Comments to the Author

You may also provide optional suggestions and comments to authors that they might find helpful in planning their study.

Reviewer #1: The abstract reads as if it is a final paper, when it only becomes obvious that it is not by the absence of a results section. This should have been made clearer at the outset. Also, it would have been good to read about some preliminary results from the pilot study.

The work plan looks reasonable, with a decent procedure and interesting question., There is a good number of participants (N=60). I have a number of minor questions, however:

Why aren't all participants interviewed?

The five studies that are referred to on p25 -31 need to be more clearly stated to remind the readers that they refer to the original objectives of p15. i.e. a clearer link.

The use of Fitbits to monitor step counting is an interesting approach. It would be helpful to elaborate on exactly how step counting is related to cooperation. In that same study, I am also curious as to whether it is cooperation vs competition that might be the driving force. (There is related work using similar wearables to measure social behaviour in autism, that it might be helpful to look into.)

Early in the manuscript the phrase, "who developed ASD during her second year of life", might be problematic for some – it makes various assumptions about the causes of autism. Perhaps, “who were diagnosed as ASD” might be more appropriate.

7. PLOS authors have the option to publish the peer review history of their article (what does this mean?). If published, this will include your full peer review and any attached files.

Reviewer #1: No

---

## [Author Response · Author response to Decision Letter 0]

6 Mar 2024

Comment: The abstract reads as if it is a final paper, when it only becomes obvious that it is not by the absence of a results section. This should have been made clearer at the outset. 

Answer: Thank you for this important comment. We revised the abstract to clearly state that this is a study protocol (on page 2 lines 19-20).

Comment: Also, it would have been good to read about some preliminary results from the pilot study.

Answer: Thank you for suggesting to include preliminary results from the pilot. We revised the Study status section to clearly state that no data was collected during the assessment of the intervention protocol (on page 26 lines 514-518), therefore there are no preliminary results.

Comment: Why aren't all participants interviewed?

Answer: Thank you for this question. We revised the Qualitative data collection section to state that interviews will be conducted till no new themes or insights emerge from additional interviews (on page 23 line 461-462).

Comment: The five studies that are referred to on p25 -31 need to be more clearly stated to remind the readers that they refer to the original objectives of p15. i.e. a clearer link.

Answer: Thank you for this important comment. We revised the Data analysis section to refer to the original objectives (on page 25 line 489-490).

Comment: The use of Fitbits to monitor step counting is an interesting approach. It would be helpful to elaborate on exactly how step counting is related to cooperation. In that same study, I am also curious as to whether it is cooperation vs competition that might be the driving force. (There is related work using similar wearables to measure social behaviour in autism, that it might be helpful to look into.)

Answer: Thank you for these important comments. We revised the Methods section to elaborate on the driving force in this specific cooperation task and to link step counting to cooperation via calculating step rate, representing the participant’s effort (on page 22-23 line 433-446).

Comment: Early in the manuscript the phrase, "who developed ASD during her second year of life", might be problematic for some – it makes various assumptions about the causes of autism. Perhaps, “who were diagnosed as ASD” might be more appropriate.

Answer: Thank you for this important comment. We revised the Introduction section as follows (on page 4 line 61).

---

## [Decision Letter · Decision Letter 1]

17 Apr 2024

PONE-D-23-06520R1The effectiveness of group interpersonal synchrony in young autistic adults’ work environment: A mixed methods RCT study protocolPLOS ONE

Dear Dr. Dvir,

Thank you for submitting your manuscript to PLOS ONE. After careful consideration, we feel that it has merit but does not fully meet PLOS ONE’s publication criteria as it currently stands. Therefore, we invite you to submit a revised version of the manuscript that addresses the points raised during the review process.

We look forward to receiving your revised manuscript.

Kind regards,

Stergios Makris

Academic Editor

PLOS ONE

Journal Requirements:

Reviewers' comments:

Reviewer's Responses to Questions

**Comments to the Author**

1. Does the manuscript provide a valid rationale for the proposed study, with clearly identified and justified research questions?

Reviewer #2: Partly

2. Is the protocol technically sound and planned in a manner that will lead to a meaningful outcome and allow testing the stated hypotheses?

Reviewer #2: Partly

3. Is the methodology feasible and described in sufficient detail to allow the work to be replicable?

Reviewer #2: Yes

4. Have the authors described where all data underlying the findings will be made available when the study is complete?

Reviewer #2: Yes

5. Is the manuscript presented in an intelligible fashion and written in standard English?

Reviewer #2: Yes

6. Review Comments to the Author

You may also provide optional suggestions and comments to authors that they might find helpful in planning their study.

Reviewer #2: What is the “mixed methods approach”?

Need to specify primary aims (confirmatory?) and secondary aims (exploratory?). The small sample size will not have sufficient power for so many hypothesis testings. The authors mentioned that the sample size will not support mediating or moderating analysis, then these analyses may not be performed.

Sample size calculation: F=0.2 needs to be justified.

Specify primary endpoint and secondary endpoints.

Line 261: “between-subject” can be omitted.

P values needs to adjusted for multiple tests.

7. PLOS authors have the option to publish the peer review history of their article (what does this mean?). If published, this will include your full peer review and any attached files.

Reviewer #2: No

---

## [Author Response · Author response to Decision Letter 1]

6 Jun 2024

What is the “mixed methods approach”?

We revised the abstract to clearly state that a convergent mixed methods design will be applied (on page 9 line 189-190):

A convergent mixed methods design will be applied, where quantitative and qualitative data are collected and analyzed in parallel.

Need to specify primary aims (confirmatory?) and secondary aims (exploratory?). 

Thank you for this important suggestion. We revised the abstract to specify one primary objective and four secondary objectives (on page 9 lines 178-186):

The primary objective is to determine whether a synchronized group intervention will have an immediate and/or long-term positive effect on participants’ prosociality and work-related stress. The secondary objectives are to determine: (1) Whether a synchronized group intervention will have an immediate and/or long-term positive effect on participants’ social closeness and sense of belonging, (2) Whether this effect will be influenced by participants’ need to belong as reported before the intervention, (3) How participants perceive the intervention as affecting their prosociality and work-related stress, and (4) In what ways the participants’ perception will contribute to a better understanding of the intervention effect.

The small sample size will not have sufficient power for so many hypothesis testings. The authors mentioned that the sample size will not support mediating or moderating analysis, then these analyses may not be performed.

Thank you for this important comment. Since we mentioned that the sample size might not support mediating analysis, we simplified the theoretical model and data analysis (on the Introduction section pages 7-8 lines 145-171, Figure 2 page 9 line 173, and the Quantitative data analysis section pages 25-26 lines 503-515). 

Sample size calculation: F=0.2 needs to be justified.

We revised the abstract to justify the selection of the effect size defined as f=0.25 for sample size calculation that supports the moderating model (on pages 11-12 lines 237-244):

A meta-analysis on prosocial consequences of interpersonal synchrony among neurotypical adults found a medium-sized effect of interpersonal synchrony on prosociality with regard to both attitudes and behavior towards individuals or groups involved in the same intervention [28]. Therefore, an effect defined as d = 0.25 may be expected [50]. An a-priori power analysis using the G*Power computer program [49] indicated that a total sample size of 40 participants would be needed to detect medium-sized effects defined as f=0.25, with 80% power and alpha at .05, using a repeated measure, within-between interaction ANOVA with four groups and three measurements.

Specify primary endpoint and secondary endpoints.

We revised the abstract to specify primary and secondary outcomes (Table 2 on page 20 line 383):

Table 2. Study outcomes

Outcomes

Primary Outcomes

 Work-related stress 

 Non-confidential behavioral cooperation 

 Confidential behavioral cooperation 

Secondary Outcomes

 Social closeness

 Friendship closeness

 Sense of belonging

Line 261: “between-subject” can be omitted.

We omitted the “between-subject” (on Page 12 line 262).

P values needs to adjusted for multiple tests.

Thank you for this important comment. We revised the abstract to state that we will use a statistical procedure to adjust P values for multiple tests (on page 25 lines 498-499):

To control the family-wise error rate when performing multiple hypothesis tests, Bonferroni Step-Down (Holm) will be applied.

---

## [Editor Report · Decision Letter 2]

16 Jul 2024

The effectiveness of group interpersonal synchrony in young autistic adults’ work environment: A mixed methods RCT study protocol

PONE-D-23-06520R2

Dear Dr. Dvir,

We’re pleased to inform you that your manuscript has been judged scientifically suitable for publication and will be formally accepted for publication once it meets all outstanding technical requirements.

Kind regards,

Stergios Makris

Academic Editor

PLOS ONE
---

## [Editor Report · Acceptance letter]

22 Jul 2024

PONE-D-23-06520R2 

PLOS ONE

Dear Dr. Dvir, 

I'm pleased to inform you that your manuscript has been deemed suitable for publication in PLOS ONE. Congratulations! Your manuscript is now being handed over to our production team.

Kind regards, 

on behalf of

Dr. Stergios Makris 

Academic Editor

PLOS ONE